# The History of *Desulfovibrio gigas* Aldehyde Oxidoreductase—A Personal View

**DOI:** 10.3390/molecules28104229

**Published:** 2023-05-22

**Authors:** José J. G. Moura

**Affiliations:** LAQV, NOVA School of Science and Technology|FCT NOVA, 2829-516 Caparica, Portugal; jose.moura@fct.unl.pt

**Keywords:** molybdenum enzymes, aldehyde oxidoreductase, sulfate-reducing bacteria

## Abstract

A story going back almost 40 years is presented in this manuscript. This is a different and more challenging way of reporting my research and I hope it will be useful to and target a wide-ranging audience. When preparing the manuscript and collecting references on the subject of this paper—aldehyde oxidoreductase from *Desulfovibrio gigas*—I felt like I was travelling back in time (and space), bringing together the people that have contributed most to this area of research. I sincerely hope that I can give my collaborators the credit they deserve. This study is not presented as a chronologic narrative but as a grouping of topics, the development of which occurred over many years.

## 1. Context

The Special Issue, “Molybdenum and Tungsten Enzymes—State of the Art in Research”, is an impressive study by Professor Ralph Mendel, and I think it is presently taking shape as the project is fully underway. The editor wanted to present a collection of historical reviews from a personal perspective by pioneers/actors in the Mo-W field. The articles, written in the first-person narrative, are supposed to present a personal account of the way the field has developed, highlighting the personal story of each author and revealing the interactions/collaborations with other key players. Professor Mendel started the series with “The History of the Molybdenum Cofactor—A Personal View” (*Molecules* 2022, 27, 4934). In this first article, we learn about the role of molybdenum (Mo) as an essential micronutrient in biology, and the way the metal is incorporated in the active center of Mo enzymes and how Mo is complexed by a pterin scaffold to form the molybdenum cofactor (Moco). My article was inspired to use this first paper as an example. 

## 2. Framing the History

My research group (in collaboration with Isabel Moura) has worked for many years on studying the fundamental role of metal ions in biology, to reveal structural–functional relationships in proteins and enzymes that are involved in crucial pathways in areas such as energy, health, agriculture, and the environment. My story begins in the 1970s and is a good example of how basic research can have an impact in an applied context. 

It all started in those early days when we observed a strange pink protein band in the chromatographic purification steps of a bacterial extract obtained from the biomass of a sulfate-reducing bacterium, *Desulfovibrio gigas* (*Dg*), a bacterium that, subsequently, was found to be involved in the accumulation of sulfide (environment) and in hydrogen production, and was also associated with methanogens (energy production). The isolated protein, after biochemical/chemical and spectroscopic studies, was eventually shown, to our great surprise, to contain molybdenum [1,2].

## 3. The Starting Point and Developments

When I started my PhD, after obtaining a degree in Chemical Engineering at the Instituto Superior Técnico, University of Lisbon (IST), JJR Fraústo da Silva was my mentor, a remarkable man, with great interests in Inorganic and Analytic Chemistry centered on chemical equilibrium and stability constants between metals and ligands. He was greatly involved in educational topics, as well as cultural issues, and at certain stages in his life, he was the Director of IST, the Minister of Education in Portugal, and the Director of the Cultural Center of Belém in Lisbon. When on sabbatical leave at the University of Oxford, he started a fruitful collaboration with RJP Williams, which can be considered as ground zero for Bioinorganic Chemistry. In the search for a PhD tutor, and following his advice, I became acquainted with AV Xavier, a former PhD student of RJP Williams, who had recently returned to Portugal. António had interests in NMR and structural paramagnetic probes and was making the first steps toward the structural determination of paramagnetic proteins by NMR. Searching for a topic for me, he met Professor Jean LeGall from CNRS-Marseille at a Conference in Paris, and I became the go-between for the Lisbon and Marseille groups and was set to learn how to purify metalloproteins from sulfate-reducing bacteria (SRB) for future spectroscopic and structural studies. As a newcomer, working in LeGall’s lab was an amazing experience. Not only did I learn a lot, but I also found that the crude extracts of the SRB were loaded with metalloproteins, and the transition metal content enabled the clear observation of colorful charge transfer bands at each chromatographic step. Therefore, we did not need to measure or follow the product’s activity, but simply its color. There was an almost artisanal quality to the manipulation of our chromatographic experiments, where large columns with manual gradients used resins, such as Sephadex, Alumina, DEAE, CMC, and others. Professor Jean LeGall also had many beautiful secrets to teach, and I almost felt like a sorcerer’s apprentice. Bacterial cells were grown in very large anaerobic containers in the Fermentation Plant at the CNRS facility, since, at that time, cloning and the overexpression of metalloenzymes were in their infancy. The clear message to all that were involved in the purification steps was that nothing was to be discarded, and we paid a lot of attention to the colored (visible) and uncolored fractions (measured at 280 nm). The procedures were time-consuming, and many hours were devoted to bench work that was sometimes difficult to interrupt. However, a fantastic pallet of colors appeared before our eyes: reds (cytochromes, rubredoxins); yellows (flavodoxins and other flavoproteins); greens (sulfite reductases); browns (for ferredoxins, hydrogenases, and APS reductases); and many others. During a very large-scale purification process (500–1000 g wet weight of cells), a strange pink fraction was observed that displayed a visible “fingerprint” of a [2Fe-2S] center found in plant-type proteins; however, to our surprise, the Mwt was quite high (around 110 KDa) and the iron content was low. Immediately, different hypotheses were proposed and, at this stage, without further characterization nor activity assigned, the “pink protein” was named by LeGall as *Zedoxin* (after my nickname Zé for José). Later, the name was changed to Mop (molybdenum protein) and then eventually to AOR, when Mo was identified, and an activity assigned. In fact, the search for the type of metal the protein contained was conducted in collaboration with JPM Cabral, at the Nuclear Research Center (he was my previous Professor at the Engineering School), where one Mo atom was revealed using thermal neutron activation analysis, in addition to 2x [2Fe-2S] centers. The initial results were presented at the 2nd Climax International Conference on the Chemistry and Uses of Molybdenum in 1976 (the first one took place in 1973) where discussions on the Inorganic Chemistry of Molybdenum occurred along with new topics, such as Bioinorganic Chemistry, including nitrate reductase and xanthine oxidase [3]. This first publication included a preliminary characterization of the “pink protein” as a molybdenum iron–sulfur protein originally isolated from *Desulfovibrio gigas*, and EPR and CD results were included to support the presence of the [2Fe-2S] clusters [1].

At this time, I had an independent research group and we moved from Centro de Química Estrutural at IST to the Gulbenkian Research Centre in Oeiras, and finally to the campus of the School of Science and Technology, NOVA, in Caparica. Over the years, many students and post-Doctorates have been involved with AOR and related research topics, namely, Belarmino Barata, Rui Duarte, Susana Andrade, Maria José Feio, Jorge Caldeira, Maria Rivas, Anders Thapper, Pablo González, Nuno Palma, Ludwig Krippahl, Luísa Maia, Jacopo Marangon, and Patrícia Paes De Sousa. Where appropriate, these collaborators are mentioned in their specific context, along with different students from other collaborating groups.

## 4. Developing Collaborations and Revealing the Properties of the Metal Sites

After the preliminary characterization, we immediately noticed that our protein was analogous to Mo-mononuclear enzymes from the xanthine oxidase (XO) family, with similarities to XO that contains a Mo center bound to a pterin moiety along with two [2Fe-2S] centers and an additional FAD lacking flavin (deflavoform). I then started a very productive collaboration with Robert (Bob) Bray´s group at the School of Biological Sciences at University of Sussex (UK). I remember him as having a very shy, curious, and critical attitude, and since he had worked for a long time with bovine milk XO, it was clear that he could immediately identify the similarities and differences between *Dg* Mop and XO. EPR was the spectroscopic tool of choice to try and detect the paramagnetic Mo(V) centre. I sent Belarmino Barata, one of my first PhD students, to Sussex to spend some time in Bob’s group. EPR signals at 9 and 35 GHz, in ^1^H_2_O and ^2^H_2_O, were obtained after the extended dithionite reduction of the protein. Using computer simulations, the parameters corresponding to those of the slow signal were obtained, such as the inactive desulfo form of various molybdenum-containing hydroxylases. Another signal obtained by a short dithionite reduction of the protein was shown, by EPR, to be of a rapid type, confirming that the protein was a molybdenum-containing hydroxylase. Activity measurements revealed that it had aldehyde: 2,6-dichlorophenol-indophenol oxido-reductase activity. No activity towards xanthine or purine was observed. Salicylaldehyde was a particularly good substrate, and a rapid signal was obtained with this substrate. The molybdenum cofactor released from the protein was active in a nit-1 *Neurosporacrassa* nitrate reductase assay. It was concluded that the protein was an aldehyde oxidase or dehydrogenase [4].

Continuing our collaboration with Sussex, X- and Q-band EPR spectra of the reduced iron–sulfur centers of *Dg* AOR were found to be consistent with the view that only two types of [2Fe-2S] clusters were present, as in eukaryotic molybdenum-containing hydroxylases [5].

At that time, as measuring mid-point redox potentials of metalloproteins was a challenge, EPR appeared as a promising way to follow the appearance and disappearance of paramagnetic redox species when the systems were poised at different potentials in the presence of redox mediators. For the oxidation–reduction studies of *Dg* AOR, the paramagnetic species to be followed were Mo(V) and the reduced [2Fe-2S] clusters. Richard Cammack introduced us to experiments, which were applied with Bob Bray to XO, where pH and temperature effects were considered. Additionally, at that time, Cammack was crucial for the development of my research group, helping us to install our first EPR instrument.

The iron–sulfur centers (I and II) could readily be distinguished by means of g-values, temperature effects, oxidation–reduction potentials, and reduction rates. Long equilibration times (30 min) with dye mediators under reducing conditions were necessary to observe the very slow equilibrating (slow-type) molybdenum signals [6].

Further characterizations of the iron–sulfur centers were performed in collaboration with BH (Vincent) Huynh at Emory in Atlanta (USA) using Mossbauer spectroscopy [7]. I met Vincent in Minneapolis at the Fresh Water Biological Institute where, after obtaining my PhD, I held a Specialist Research position, where an extended collaboration was developed with him and Eckard Munck. I was then invited as Adjunct Associated Professor to UGA, only 70 miles away from Atlanta; thus, we maintained a strong collaboration over a wide range of topics: ferredoxins, rubredoxin-type proteins, hydrogenase, and sulfite reductase. Mossbauer studies of molybdenum hydroxylases prior to this work were performed on a natural abundance sample of milk XO by Russ Hille and collaborators [8]. We grew ^57^Fe-enriched *Dg* cells (an advantage of using bacteria compared to mammalian systems) and purified AOR samples that enabled iron–sulfur center site-specific labeling with enhanced sensitivity for Mossbauer studies. The spectra of the enzyme in its oxidized, partially reduced, and benzaldehyde-reacted states were recorded at different temperatures, and applied magnetic fields clearly supported the view that all the iron atoms were organized in [2Fe-2S] clusters. With a strong homology related to plant-type ferredoxins, in the oxidized enzyme, the clusters were diamagnetic. Mossbauer spectra of the reduced clusters also showed the characteristics of a spin-coupling model proposed for a [2Fe-2S] cluster, where a high-spin ferrous ion was antiferromagnetically coupled to a high-spin ferric ion. Two ferrous sites were observed for the reduced enzyme, indicating the presence of two types of [2Fe-2S] clusters in the *Dg* AOR [7]. Taking this observation together with the re-evaluated value of the iron content (3.5 ± 0.1 Fe/molecule), we concluded that, similar to other Mo-hydroxylases, *Dg* AOR also contained two spectroscopically distinguishable [2Fe-2S] clusters.

Another Emory UGA connection was established with Kwok To Yue, to perform resonance Raman studies on the iron–sulfur centers of *Dg* AOR, which provided further insights into the iron clusters and their relation to the protein environment [9].

Previously, the molybdenum site of *Dg* AOR was scrutinized by EXAFS using fluorescence detection and synchrotron radiation in collaboration with Graham George [10]. In the oxidized form, the molybdenum environment was found to contain two terminal oxo groups and two long (2.47 A) Mo-S bonds, as expected. The behavior of both oxidized and dithionite-treated forms was found to be similar to that observed previously with “desulfo” XO.

## 5. The 3D Structure We Were Waiting For!

In 1995, when the Dg AOR structure was finally solved, many novel molecular arrangements involved in some of the most primordial processes on Earth were revealed for the first time. These processes include respiration and O_2_ utilization, the biocycles of elements such as S and N, nitrogen utilization, biofuels (hydrogen), the involvement of Mo and W in Biology, and processes crucial for the understanding of the origin of life. I was so excited with all this new information that I decided to write a short article on all the structures we wanted to determine and those expected to be discovered [11]. It really was a vintage year.

In the same year, the structures of cytochrome-*c* oxidase [12,13], di-heme cytochrome-*c* peroxidase [14], cytochrome-*cd*_1_ nitrite reductase [15], urease [16], sulfite reductase [17], nickel–iron hydrogenase [18], ribonucleotide reductase [19], aldehyde oxidase [20], purple acid phosphatase [21], DMSO reductase [22], and formate dehydrogenase [23] were solved, with the last two being solved in 1996.

The impact of the newly discovered structures was so impressive that a series of scientific notes were published highlighting the results using catchy titles, such as “The Structure of Cytochrome c Oxidase, Energy Generator of Aerobic Life”, “The Purpose of Proton Pathways”, “Splitting Molecular Hydrogen”, and “At Last-The Crystal Structure of Urease” [24,25,26,27].

When we needed to sequence and structurally characterize *Dg* AOR, as a representative of mononuclear Mo-enzymes that were closely related to XO, a collaboration with Prof. Claudina Rodrigues Pousada from the Gulbenkian Research Institute at Oeiras, was initiated. Orfeu Flores, a previous Chemistry student of mine, was sent to her laboratory to perform the molecular cloning and sequence analysis of the gene of the molybdenum-containing *Dg* AOR and infer from the deduced amino acid sequence similarities with XO. Jozef J. Van Beemen and co-workers (University of Ghent, Belgium) were also involved at this point via a previous collaboration. The protein sequence showed a 52% (on average) similarity to xanthine dehydrogenases from different organisms. The codon usage of aldehyde oxidoreductase was almost identical to the calculated codon usage for the *Desulfovibrio* bacteria [28]. Incidentally, Orfeu Flores continued with his molecular biology studies and eventually obtained the sequence of the whole *Desulfovibrio gigas* genome [29].

The project gained further “élan” when Maria João Romão (my ex-student at IST enrolled in a Chemical Engineering degree) finished her PhD in Organic Chemistry and was searching for a project to initiate a post-Doctoral thesis with Robert Huber at the Max Planck Institut fur Biochemie, Martinsried, Germany. I remember traveling to visit Robert Huber to talk about the state of our knowledge on this enzyme, which resembled deflavo-XO, and where we had extensive discussions on the multiple coordination possibilities for the Mo site.

The structure of *Dg* AOR was solved at a 2.25 A resolution with the overall 3D-folding determined and structural details on the active site reported. The Mo cofactor was defined as an MCD, and the structure was the first XO family member for which a crystal structure had been determined [30,31]. Later, a high-resolution structure was obtained (1.28 A) with amazing definition for the metal centers and for the pterin rings [32]. It was evident that *Dg* AOR was structurally similar to mammalian XO, containing an MCD pterin moiety (different from XO) and only two Fe/S centers, but lacked the flavin-binding-site domain (see Figure 1). I stress that this analogy is important for further developments on electron transfer to other proteins and how to complement the lack of a flavin site.

The ligands around the Mo were completely determined and a water molecule in the vicinity of the metal site was identified as that involved in O-atom insertion in the substrate (conversion of aldehyde to carboxylic acid). Inhibitors were also studied, identifying the Mo site as a drug target. The protein molecule is folded into four domains (the first two bind the iron–sulfur clusters and the others the Moco). The Mo site is deeply buried in the protein; although, it is accessible through a well-defined tunnel. The molybdenum is penta-coordinated with two dithiolene sulfur atoms of one molybdopterin (the cofactor identified as pyronopterin cytosine dinucleotide) and three oxygen ligands, one of which is presumably an oxo or a sulfide group in the functional sulfo-form of the enzyme analogous to XO [31,32]. The bacterial structure was used extensively as a model for human XO until its structure was finally solved in 2000 [33].

The 3D structure of the XO-related molybdenum-iron protein *Dg* AOR was analyzed in its desulfo-, sulfo-, oxidized, reduced, and alcohol-bound forms at different resolutions by Huber et al. [33]. A remarkable finding was the presence of a bound isopropanol inhibitor in the inner compartment of the substrate-binding tunnel, which was the inspiration for the mechanistic details of and reaction to aldehydes. The occupancy of the substrate-binding site by the isopropanol (an inhibitor), the observation of a buried water molecule as a source of O-transfer, and the conserved Glu 869 involved in catalysis were all crucial for the proposal of a mechanism. I include here (Figure 2) the first proposal for a complex with an aldehyde substrate close to Mo(VI), the enzyme/carboxylic acid product complex (Mo(IV)) and the intermediate, after product dissociation, with Glu 869 bound to Mo.

The reductive half-cycle of the hydroxylation reaction of AOR was also extended to XO [34,35,36,37]. Our hypothetical structures depicted: (a) the Michaelis–Menten complex with Mo(VI) and aldehyde substrate; (b) the enzyme carboxylic acid product complex with Mo(IV); and (c) an intermediate after product dissociation, in which Glu869 is bound to the metal, supported by an increased electron density between the dithiolene sulfur atoms in the oxidized state and decreased electron density in the reduced state. The active site’s multiple forms are presented in Figure 3. The presence of desulfo forms (slow EPR signals) was always a central argument of the discussion. Robert Huber [34] attempted some resulfuration procedures; however, extra added sulfur was observed to occupy an apical and not the equatorial position, as observed in functional XO. *Dg* AOR, as a member of the XO family, was accepted as an exception with a catalytically competent form having an equatorial oxo ligand instead of a sulfido ligand.

The complete catalytic mechanism of xanthine oxidase was latterly determined using a computational study (computational QM/MM approach) in collaboration with Nuno Cerqueira, Henrique Fernandes, and Luísa Maia [38]. 

In relating all these results, I remember Rui Duarte, my PhD student, for numerous contributions and support.

## 6. After the Structure: Further Insights into Catalysis, Inhibition, and Reactivity

After the high-resolution structure was determined, several further studies were undertaken (spectroscopic and structural), and I had the pleasure of hosting Carlos Brondino as a visiting Professor from Santa Fé, Argentina, in my laboratory for a few years, and two PhD students, Pablo Gonzalez and Maria Rivas, who, after obtaining their PhD degrees with us, obtained post-Doctoral positions where they contributed with their expertise in EPR and Biochemistry. Combining spectroscopy and X-ray crystallography, a collaboration with Maria João Romão’s group (with contributions from Teresa Santos Silva, Roeland Boer, and Hugo Correia), provided more detail on inhibitor steady-state kinetics and reactions with cyanide, ethylene glycol, and glycerol as reversible inhibitors. Kinetic data with both cyanide and samples prepared from single crystals confirmed that *Dg* AOR did not need a sulfido ligand for catalysis and confirmed the absence of this ligand in the coordination sphere of the molybdenum atom in the active enzyme [39,40].

The X-ray structure of the ethylene glycol and glycerol-inhibited enzyme, where the catalytically labile OH/OH_2_ ligand is lost, defined the coordination mode for both alcohols. These two adducts showed a direct interaction between the molybdenum ion and one of the carbon atoms of the alcohol moiety, constituting the first structural evidence for such a bond in a biological system [40].

Jacopo Maragon, an Italian PhD student, joined the group to study the kinetic and structural aspects of *Dg* AOR, and revealed a dithiolene-based chemistry for enzyme activation and inhibition by H_2_O_2_. Incubation with dithionite plus sulfide in the presence of dioxygen produces hydrogen peroxide not associated with enzyme activation. The peroxide molecule coordinates to molybdenum in a η^2^ fashion inhibiting enzyme activity. The fact that *Dg* AOR does not need a sulfido ligand for catalysis, indicates that the process leading to the activation of inactive *Dg* AOR samples is different to that of desulfo-XO [41].

I had the opportunity to further probe inhibition and reactivity behaviors toward aldehydes. I found that arsenite, an inhibitor, interferes at the position of substrate binding and the coordination mode of the arsenite suggests that the substrate reacts with a labile water ligand of the Mo site to form a Mo–O–C bond instead of a Mo–C bond. I also found that the reactivity and molecular details of the enzyme–substrate and enzyme–product interactions could be probed using kinetic studies, showing that *Dg* AOR catalyzes the oxidative hydroxylation of aromatic aldehydes, but not heterocyclic compounds. A most important result was achieved using the NMR spectroscopical studies of ^13^C-labeled benzaldehyde that confirmed that *Dg* AOR can catalyze the conversion of aldehydes to their respective carboxylic acids [42,43].

The 3D structure also provided information about the surface and charge distribution of the enzyme. The direct electrochemistry of *Dg* AOR detected different orientations of the enzyme towards the surfaces of electrodes, and the voltametric behavior of the enzyme was analyzed using gold and carbon electrodes (pyrolytic graphite and glassy carbon). This was a continued collaboration with Margarida Correia dos Santos, Patrícia Sousa, and M. Lurdes Goncalves (Engineering School, Lisbon) [44]. Two different strategies were used: one with the molecules confined to the electrode surface and a second with *Dg* AOR in solution. In all the cases studied, electron transfer occurred, although different redox reactions were responsible for the voltametric signal. From a thorough analysis of the voltametric responses and structural properties of the molecular surface of *Dg* AOR, the redox reaction at the carbon electrodes could be assigned to the reduction in the more exposed iron cluster, [2Fe-2S] II, whereas the reduction in the molybdopterin cofactor occurred at the gold electrode, and catalytic currents were observed. The voltametric results in the presence of aldehydes were also reported and discussed.

## 7. Electron-Transfer Complexes

I have been long-interested in studying protein–protein interactions using different methodologies, namely, ^1^H and ^13^C NMR, especially for the study of binary complexes, such as the interaction of multiheme cytochromes with rubredoxin, flavodoxin, ferredoxin, and FNR, and for the measurement of specific catalytic currents using electrochemistry. Other systems of interest included electron transfer from substrates to electrodes mediated by enzymes and small-electron-carrier proteins (i.e., peroxidases and monoheme cytochromes) and cross-linking methodologies. Previously, we accumulated evidence of the complex formation between flavodoxin and cytochrome c_3_ using ^1^H-NMR and molecular modeling studies (Cristina Correia in my laboratory and Enrico Monzan, visiting us from University of Pavia, Italy) [45].

At this point, I was very aware that bacterial AORs from *Desulfovibrio* species had a slightly different molybdenum center, with a second equatorial oxo group in place of the more frequent sulfo group, and it harbored only two Fe/S centers, apart from the molybdenum center (no FAD center). Despite this, when the AOR structure is represented with its putative physiological partner, the flavin-containing flavodoxin, which can be regarded as a pseudo subunit, it becomes apparent that the structural homology with XO is preserved (as indicated in Figure 1).

Keeping in mind the close relation to deflavo-XO, I searched for proteins, such as flavodoxin, which could act as a missing link in electron-transfer processes. ET transfer from aldehydes (AORs) to H_2_ (hydrogenase) through flavodoxin and cytochrome c_3_ was successfully tested. In vitro, we reconstituted an electron-transfer chain from aldehydes to the production of molecular hydrogen (Figure 4).

AOR was shown to be part of an electron-transfer chain comprising four different soluble proteins from *Dg,* with a total of 12 discrete redox centers, which can link the oxidation of aldehydes to the reduction of protons [4,46].

At that point in time, two excellent PhD students (Nuno Palma and Ludwig Krippahl) developed a new computationally efficient and automated “soft-docking” algorithm to assist the prediction of the mode of binding between two proteins, using the three-dimensional structures of the unbound molecules. The method was implemented in a software package called BiGGER (Bimolecular Complex Generation with Global Evaluation and Ranking) [47,48].

This approach was used to model the electron-transfer complex between *Dg* AOR and flavodoxin, in search of the “missing” flavo-domain. Three-dimensional protein structures of the XO family showed different solutions for the construction of a “wire” for transferring electrons between the flavin adenine dinucleotide (FAD) group and molybdenum cofactor. As case studies, in XO, all the cofactors lay within the domains of the same protein chain, whereas in CO dehydrogenase, the Fe–S centers, FAD, and Mo cofactors, were enclosed in separate chains and the enzyme existed as a stable complex of all three (Figure 5). In aldehyde oxidoreductase, only Fe–S and Mo co-factors were present in a single protein chain. Flavodoxin was docked to aldehyde oxidoreductase to mimic the flavin component of the intramolecular electron-transfer chains of XO and CO dehydrogenase and, remarkably, the main features of the electron-transfer pathway were observed [49].

## 8. Other AORs Isolated from Sulfate Reducers

Aldehyde oxido-reductase activity has also been observed in different sulfate-reducing organisms: in *Desulfovibrio (D.) desulfuricans* ATCC 27774, a sulfate reducer that can use nitrate as an alternative respiratory substrate to sulfate; in *D. alaskensis* NCIMB 13491, a strain isolated from a soured oil reservoir; and in *D. aminophilus*, an aminolytic strain performing thiosulfate dismutation [50,51,52,53].

There is evidence of strong homology between the three enzymes, and *Dg* AOR, in terms of the biochemical data, substrate utilization, visible and EPR Mo(V), depicted a hyperfine interaction with one proton associated with MoV species after prolonged reduction as well as the presence of two distinct [2Fe-2S] clusters. The gene sequence and crystal structure of AOR from *D. desulfuricans* ATCC 27774 indicated an overall fold very similar to *Dg* AOR. For *D. alaskensis* AOR, the assignment of the proximal [2Fe-2S] cluster to the Mo site was performed, correlating spectroscopic and structural data, providing further evidence to support the interactions among paramagnetic centers (Mo(V) and EPR-distinguishable Fe/S I and Fe/S II), which is explored further (see below).

## 9. Magnetic Interaction between the Redox Centers of AOR (and XO)

With the high-resolution structure of *Dg* AOR in hand, I had a good system to further explore the magnetic interactions within the redox centers, as well as the structural/spectroscopic assignments and electronic pathways. Two important collaborations were established with Carlos Brondino and Pablo Gonzalez (and A. C. Rizzi and co-workers from Argentina) who contributed to a topic of common interest: magnetic interactions.

A series of articles were published on the isotropic exchange interaction between Mo and the proximal FeS center in the XO family member *Dg* AOR, for native and polyalcohol inhibited samples. EPR/QM/MM studies were completed [54] and further EPR studies treated the Mo site as a system containing weakly coupled paramagnetic redox centers with different relaxation rates [55].

Meanwhile, while being in Marseille on sabbatical leave, I had the opportunity to contact Bruno Gigliarelli and Patrick Bertrand, and Jorge Caldeira (my student) was the mediator. They performed the analyses of the EPR properties of reduced [2Fe-2S] centers in *Dg* AOR and in milk XO, assigning the Fe-S site closest to the molybdenum cofactor [56]. The studies were further extended to the identification of the reducible sites of the [2Fe-2S] clusters and, when combined with the electron-transfer pathways proposed based on the X-ray crystal structure, a detailed description of the electron-transfer system of *D*g AOR was proposed [57].

## 10. Catalysis in Non-Aqueous Solvents

I maintained a long-term collaboration with Professor Andrey Levashov from the Cryoenzymology Deptartment at the Lemonosov University in Moscow, and several visits occurred. NiFe hydrogenases were the first topic we addressed, and we collaborated via Sergey Nametkin, Levashov´s student. I really enjoy the friendship, science, and culture that I was exposed to, for several years, parallel to the undergoing political transitions. Two students, firstly Belarmino Barata and subsequently Susana Andrade, collaborated with Elvira Kamenskaya in Moscow and managed to encapsulate *Dg* AOR in reverse micelles of sodium bis-(2-ethylhexyl) sulfosuccinate in isooctane and follow the catalysis using benzaldehyde, octaldehyde, and decylaldehyde as substrates. I was quite excited by the results where the catalysis of a non-water-soluble aldehyde was achieved by an enzyme in water. I learned that, by ultrasedimentation analysis, the micelles had a 100 kDa molecular weight species, confirming the subunits could dissociate and stay active, and that the protein fold and metal cofactor could be kept intact upon encapsulation. Furthermore, the enzymatic activities for octaldehyde and decylaldehyde were detected only in reverse micelles, an exciting result. EPR studies using spin-labeled reverse micelles indicated that octaldehyde and benzaldehyde intercalated in the micelle membrane opening new possibilities for studying enzymes and the bioconversions of non-polar compounds [58].

## 11. Oxygen Atom Insertion vs. Abstraction

Molybdenum enzymes are involved in a wide range of metabolic pathways. Oxygen atom insertion, as depicted in the conversion of aldehydes to carboxylic acids, has been extensively studied for the XO family. I became involved in a new aspect, the participation of mononuclear Mo enzymes in metabolic pathways that may utilize nitrite to produce NO, a powerful signaling molecule, under anoxia. Therefore, molybdenum enzymes were described as “non-dedicated” nitrite reductases, providing mechanistic pathways for *O-atom abstraction*, an alternative to O-atom insertion (as depicted in the conversion of aldehydes to carboxylic acids). My long-term co-worker, Luísa Maia, performed very exciting contributions in the field publishing an extensive and comprehensive review of how biology handles nitrite [59], and compared nitrite utilization by bacterial AOR and human and rat XO enzymes, firmly establishing a new class of nitric oxide-forming nitrite reductases [59,60,61,62]. Xanthine oxidoreductase and aldehyde oxidase were highlighted on the figurative NO metabolism map. These included other mononuclear enzymes from different families: mammalian XD; XO and AO [61,62]; bacterial aldehyde oxidoreductase [62]; mammalian sulfite oxidase [63]; mammalian mARC [64]; plant nitrate reductase [65]; and bacterial “respiratory” nitrate reductase [66]. These new discoveries have provided pharmacological opportunities to modulate NO responses under hypoxic conditions.

## 12. From Basic to Applied Research: From Bacteria to Humans

The development of the *Dg* AOR project and investigation of the mechanisms of molybdenum-containing enzymes was very rewarding and considerably enhanced by the crystallographic determination of the high-resolution 3D structure of *Dg* AOR, the first structure of a Mo enzyme of the XO family in 1995 (28,29). It was an important moment that allowed us to overcome the lack of fine structural information on the mononuclear Mo-containing enzymes that had prevented the detailed understanding of the mechanisms operating in these enzymes for many years. The enzyme, belonging to the XO family, was a potential model for the mechanistic studies on this group of enzymes and almost simultaneously, after the determination of the first 3D structure of Dg AOR, the XO structure was determined in 1996. This structural determination of *Rhodobacter sphaeroides* DMSO reductase allowed for a direct comparison between molybdo- and tungsto-enzymes, and showed how these apparent different enzymes are surprisingly related [22].

One molybdenum-dependent metabolic process is purine catabolism. XO is responsible for catalyzing the sequential hydroxylation of hypoxanthine generally producing urate, which, in humans, is the terminal product of purine catabolism. A high concentration of urate in the blood is a risk factor for gout (deposition of monosodium urate crystals in and around the joints) and the development of many other diseases (i.e., inflammatory arthritis). Major risk factors for gout include hyperuricemia, genetics, dietary factors, among others. It is reported that the prevalence of gout worldwide ranges from 0.1% to approximately 10%, and the incidence ranges from 0.3 to 6 cases per 1000 person years [67].

Different strategies to combat the disease include increasing excretion and/or decreasing the formation of urate by inhibiting XO. For this reason, human XO is one of the targets of therapies against hyperuricemia and gout. The full characterization of Dg AOR (biochemical and first structural data) has been a fundamental achievement during a period that experienced a lack of information for related human enzymes (XO and aldehyde oxidase), which has enabled the proposed mechanisms and inhibitory sites to be studied for future drug developments (Figure 6) [35,68,69].

## 13. Epilogue

Looking back over this period of almost 40 years, the journey has revealed the new findings, new discoveries, and new steps that have positioned *Dg* AOR as a member of the of mononuclear Mo-enzyme family in terms of structure, function, and mechanism. The path that led to this discovery enabled me to meet with, collaborate with, and befriend many people from many areas of the world. It has been an affair to remember! The history of *Dg* AOR follows and parallels MOTEC reunions (Molybdenum Tungsten Enzyme Conferences) that began in Brighton, UK, in 1997, with the most recent event occurring in 2022 in Indianapolis, USA, passing through Sintra, Portugal, in 2013.

## Figures and Tables

**Figure 1 molecules-28-04229-f001:**
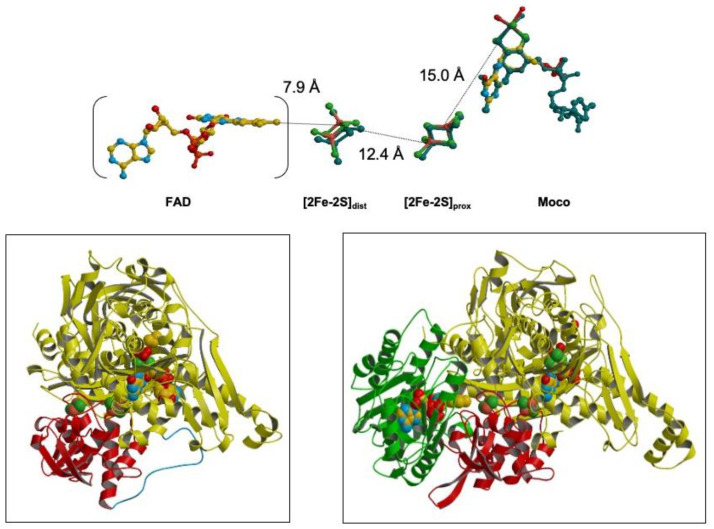
Comparison of xanthine oxidase (**right**) and *Desulfovibrio gigas* aldehyde oxidoreductase (**left**). XO contains four redox centers: Moco (the pterin cofactor is a molybdopterin cytosine dinucleotide), two [2Fe-2S] centers, and an FAD moiety. The polypeptide chain folds into distinct domains that bind the two iron–sulfur clusters and the Moco. AOR presents strong similarities to XO. The additional domain (in green) that binds FAD in XO is absent in *Dg* AOR (does not contain a flavin moiety); although, it was recognized that a small flavodoxin participates in electron transfer instead, as discussed throughout the manuscript. At the top of the figure, the distances between the redox centers are shown [31,32].

**Figure 2 molecules-28-04229-f002:**
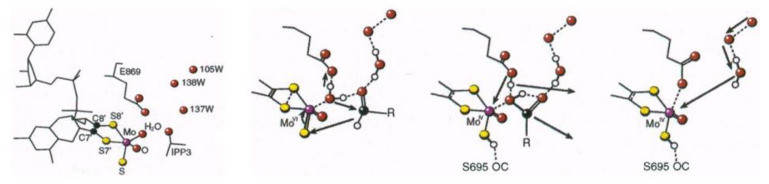
Proposed mechanism for the conversion of aldehydes in carboxylic acids based on *Dg* aldehyde oxidoreductase structural studies. An historic perspective (see text) (figure from ref. [34]).

**Figure 3 molecules-28-04229-f003:**
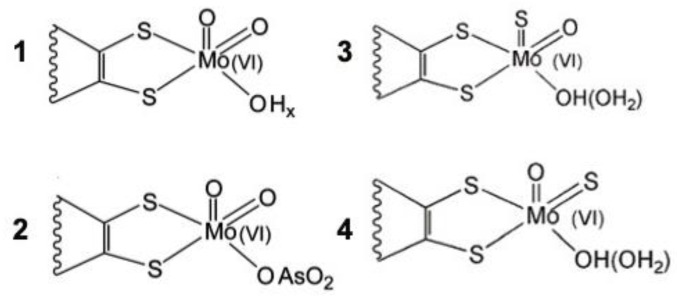
Relevant structures of the Mo site of *Dg* AOR in relation to XO. 1. *Dg* AOR active state depicting O apical coordination (desulfoform); 2. arsenite-inhibited structure; 3. resulfuration of *Dg* AOR with apical S coordination; and 4. native XO or resulfurated desulfo-XO (see text).

**Figure 4 molecules-28-04229-f004:**
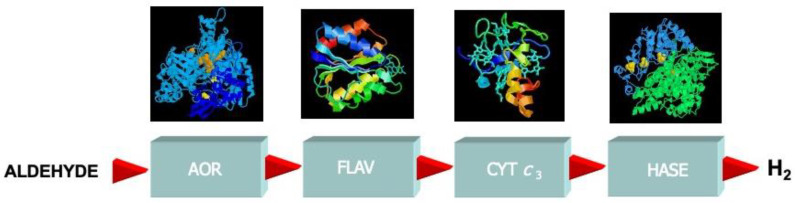
Dg AOR as part of an electron-transfer chain comprising four different soluble proteins from *Dg*, with a total of 12 discrete redox centers, capable of linking the oxidation of aldehydes to the reduction of protons (H_2_ production): AOR (MPT, 2x[2Fe-2S] clusters), flavodoxin (FMN), cytochrome *c*_3_ (4 hemes), and NiFe hydrogenase ([NiFe] cluster, 1x[3Fe-4S] and 2x[4Fe-4S] clusters).

**Figure 5 molecules-28-04229-f005:**
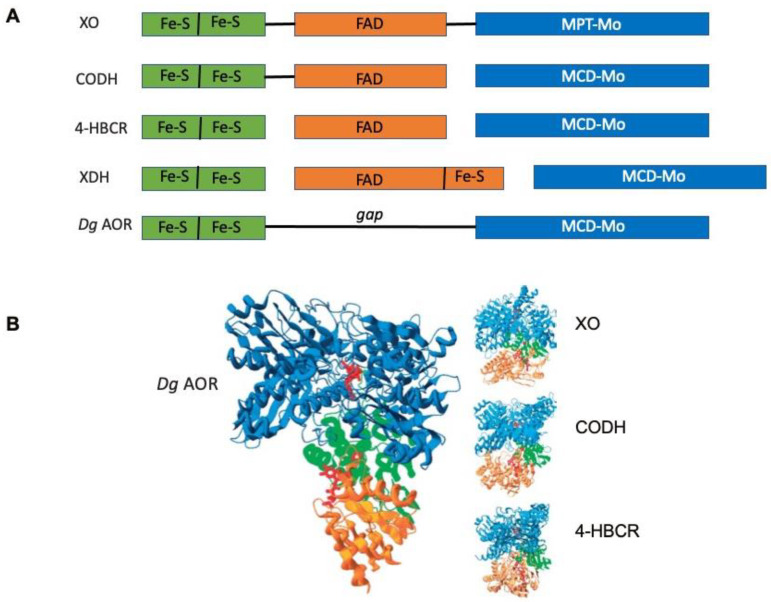
(**A**) Schematic representation of the domain sequence for the three enzymes XO, CODH, 4-HBCR (including an extra elongation of the β-subunit that binds, in addition to the FAD moiety, the [4Fe-4S] center), and AOR (homologies among molybdenum hydroxylases). The FAD domain is absent in the *D. gigas* aldehyde oxidoreductase. In *R. capsulatus* xanthine dehydrogenase and *Bos Taurus* XO, the iron–sulfur- and flavin-binding portions of the protein constitute one subunit (XdhA), and the molybdenum-binding portion a second (XdhB). In *O. carboxydovorans* CO dehydrogenase, the iron–sulfur centers are together in one subunit (CoxS), the flavin in a second (CoxM), and the molybdenum in a third (CoxL). A comparison with the *R. capsulatus* XDH domain sequence is also included (a variation in the XO domain theme). (**B**) Docking configuration for *Dg* flavodoxin and *Dg* AOR (**left** panel) compared with the structures of XO (**top right**), COD (**center right**), and 4-HBCR (**bottom right**). The colors correspond to the sequence homology schemes shown in Figure 1: green for FeS, orange for FAD and flavodoxin, and blue for MCD-Mo or MPT-Mo regions (adapted from [49]).

**Figure 6 molecules-28-04229-f006:**
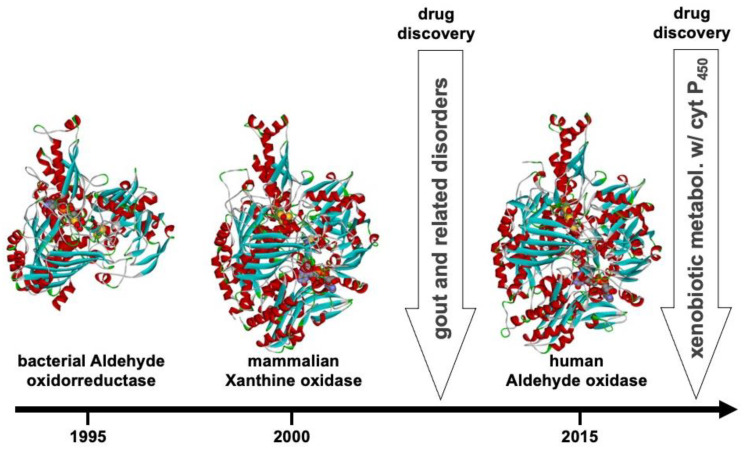
Three-dimensional structures solved on the indicated time scale: Dg AOR [1995], mammalian XO [2000], and human aldehyde oxidase [2015]. Between 2000 and 2015, numerous new structures of native and inhibited bacterial and mammalian enzymes have been determined in mechanistic studies and drug discovery research.

## Data Availability

Not applicable.

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
