# Peer review of "The History of Desulfovibrio gigas Aldehyde Oxidoreductase—A Personal View"

_molecules, 2023, doi:10.3390/molecules28104229_

Round 1

Reviewer 1 Report

This is a very nice, easy to read personal history of one of the foundational enzymes in the Mo/W field.  The author provides important historical context and conscientiously incorporates the contributions of others along the way.

Author Response

Comment: This is a very nice, easy to read personal history of one of the foundational enzymes in the Mo/W field.  The author provides important historical context and conscientiously incorporates the contributions of others along the way.

Ans: I realy appreciate the positive comments sent by the reviewer.

Reviewer 2 Report

This is a nice and comprehensive review about the history of MOP. The only part I was missing are more details about the solving of the Crystal structure of MOP, since this was a very important contribution to the field, which gave great insights into the structure of molybdoenzymes, in particular the XO family. It would be great to get more details, how the enzyme was purified and sulfurated and how long it took to get crystals and solve the structure. In particular the sulfuration of the enzyme would be helpful, since nowadays, new data were published about the mechanism not involving the sulfido-ligand. More insights would be great, also to clarify some controversies in the field. 

Author Response

Comment: This is a nice and comprehensive review about the history of MOP. The only part I was missing are more details about the solving of the Crystal structure of MOP, since this was a very important contribution to the field, which gave great insights into the structure of molybdoenzymes, in particular the XO family. It would be great to get more details, how the enzyme was purified and sulfurated and how long it took to get crystals and solve the structure. In particular the sulfuration of the enzyme would be helpful, since nowadays, new data were published about the mechanism not involving the sulfido-ligand. More insights would be great, also to clarify some controversies in the field. 

Ans: As asked by the Editor, this is an account on my journey on AOR research and all-important events were included. The fine details of each study can be found in the respective papers, all properly cited, and it is my opinion that they are out of the scope of this account. I try to tell a comprehensive story that goes back a long time period, reporting my research on Aldehyde oxidoreductase from Desulfovibrio gigas.

The development of the research involved in many aspects, using many complementary tools not so much on a chronologic narrative but grouping of topics. As strongly stated in the text, and pointed by the reviewer, the solving of the Crystal structure of MOP, was “a very important contribution to the field, which gave great insights into the structure of molybdoenzymes, in particular the XO family.”

Huber´s contribution was crucial (with Maria J. Romão, at that time his Pos-Doc). The data on crystallization and the solving the structure would be to detail and out of the scope of the review and indicated in the original articles, where details can be found.

In parallel, other methodologies used along the research work are also only briefly mentioned not going into detail.

Romão MJ, Barata BAS, Archer M, Lobeck K, Moura I, Carrondo MA, LeGall J, Lottspeich F, Huber R, Moura JJG. Eur J Biochem 1993 215, 729-32. ii) Romão MJ, Archer M, Moura I, Moura JJG, LeGall J, Engh R, Schneider M, Hof P, Huber R. Science 1995 270, 1170-6.

Rebelo J, Dias J, Huber R, Moura J, Romão M. J Biol Inorg Chem 2001 6, 791-800.

Huber R, Hof P, Duarte RO, Moura JJG, Moura I, Liu M-, LeGall J, Hille R, Archer M, Romão MJ. Proc Natl Acad Sci USA 1996 93, 8846-51.

We discuss the active site multiple forms of AOR (and XO), and the   presence of desulfoforms (slow EPR signals) was always a center piece of the discussion. Resulfuration procedures, but extra added sulfur was observed to occupy an apical and not the equatorial position, as observed in functional XO. Dg AOR, as a member of XO family, was accepted as an exception with a catalytically competent form having an equatorial oxo ligand instead of the sulfido ligand. Later, details on inhibitor steady-state kinetics and reactions with cyanide, ethylene glycol, and glycerol as reversible inhibitors, supported by kinetic data with both cyanide and samples prepared from single crystals confirmed that Dg AOR does not need a sulfido ligand for catalysis and confirmed the absence of this ligand in the coordination sphere of the molybdenum atom in the active enzyme. Additional studies using kinetic and structural arguments revealed a dithiolene-based chemistry for enzyme activation and inhibition by H2O2 and the fact that Dg AOR does not need a sulfido ligand for catalysis indicates that the process leading to the activation of inactive Dg AOR samples is different to that of desulfo-XO. These are the insights we explain in the text.

Correia HD, Marangon J, Brondino CD, Moura JJG, Romão MJ, González PJ, Santos-Silva T. J Biol Inorg Chem 2015 20, 219–229.

Marangon J, Correia HD, Brondino CD, Moura JJG, Romão MJ, González PJ, Santos-Silva T. PLoS ONE 2013 8 (12) e83234.

Boer DR, Thapper A, Brondino CD, Romão MJ, Moura JJG. J Am Chem Soc. 2004 126, 8614-5 and Thapper A, Boer DR, Brondino CD, Moura JJG, Romão MJ. J Biol Inorg Chem 2007 12, 353-66.